# Evaluation of surface-based hippocampal registration using ground-truth subfield definitions

Jordan DeKraker[1]*, Nicola Palomero-Gallagher[2,3], Olga Kedo[2], Neda Ladbon-Bernasconi[1], Sascha EA Muenzing[2], Markus Axer[2], Katrin Amunts[2,3], Ali R Khan[4], Boris C Bernhardt[1†], Alan C Evans[1†]

[1]Montreal Neurological Institute and Hospital, McGill University, Montreal, Canada; [2]Institute of Neuroscience and Medicine INM-1, Research Centre Jülich, Jülich, Germany; [3]C. & O. Vogt Institute for Brain Research, University Hospital Düsseldorf, Heinrich-Heine-University, Düsseldorf, Germany; [4]Robarts Research Institute, University of Western Ontario, London, Canada

**\*For correspondence:**
jordan.dekraker@mail.mcgill.ca

[†]co-senior authors

**Competing interest:** The authors declare that no competing interests exist.

**Abstract** The hippocampus is an archicortical structure, consisting of subfields with unique circuits. Understanding its microstructure, as proxied by these subfields, can improve our mechanistic understanding of learning and memory and has clinical potential for several neurological disorders. One prominent issue is how to parcellate, register, or retrieve homologous points between two hippocampi with grossly different morphologies. Here, we present a surface-based registration method that solves this issue in a contrast-agnostic, topology-preserving manner. Specifically, the entire hippocampus is first analytically unfolded, and then samples are registered in 2D unfolded space based on thickness, curvature, and gyrification. We demonstrate this method in seven 3D histology samples and show superior alignment with respect to subfields using this method over more conventional registration approaches.

## eLife assessment

This paper presents an **important** contribution to the field of hippocampal registration by introducing a novel surface-based approach that utilizes the topological and morphological features of the hippocampus for anatomical registration across individuals, rather than volumetric-based methods commonly used in the literature. The study provides **compelling** evidence for the efficacy of this approach using histological samples from three different datasets and offers validation of the method through comparison with traditional volumetric registration. This is **important** work given the large number of studies that examine hippocampal shape, thickness, and function in large cohorts, providing strong support for the use of hippocampal unfolding methods in future studies.

## Introduction

The hippocampus is part of the archicortex that, like the neocortex, can be further parcellated into subfields according to its cytoarchitecture (*Ding and Van Hoesen, 2015*; *Duvernoy et al., 2013*; *Palomero-Gallagher et al., 2020*). The study of hippocampal subfields is promising for both basic science since their microcircuits are thought to be fundamental to memory processes (*Milner et al., 1968*; *O'Keefe and Nadel, 1978*; *Riphagen et al., 2020*) and for the pathogenesis of several brain disorders given their vulnerabilities to many conditions, notably epilepsy (*Bernhardt et al., 2015*; *Bernhardt et al., 2016*; *Blumcke et al., 2013*; *Thom, 2014*), Alzheimer's disease (*Braak and Del*

*Tredici, 2015*; *de Flores et al., 2015*), and schizophrenia (*Haukvik et al., 2018*; *Roeske et al., 2020*). However, subfield parcellation is challenging. This relates to variability across segmentation protocols both at the level of histology and imaging (*Wisse et al., 2017*; *Yushkevich et al., 2015a*). Moreover, the hippocampus has a complex shape that varies between individuals (*Chang et al., 2018*; *DeKraker et al., 2021*; *Ding and Van Hoesen, 2015*; *Palomero-Gallagher et al., 2020*). This topic has received widespread attention, leading to the development of an international harmonization effort that focuses on extracting geometric regularities from reference histology slices that can be applied to MRI, mostly using coronal slices (*Olsen et al., 2019*; *Wisse et al., 2017*; *Yushkevich et al., 2015a*). Building on this discussion, we assert that the most basic geometric consistency of the hippocampus is a 3D folded surface and so subfield parcellation schemes should be applied using surface-based registration, similar to state-of-the-art neocortical parcellation.

Surface-based registration, either in the hippocampus or neocortex, aims to account for inter-individual differences in folding or gyrification patterns during registration (*DeKraker et al., 2021*; *Fischl et al., 2008*; *Im et al., 2008*; *Lyttelton et al., 2007*; *MacDonald et al., 2000*; *Van Essen et al., 2000*; *Robbins, 2003*). Briefly, this consists of first generating a 3D model of the structure of interest and representing it as a surface, typically a mid-thickness surface. In the neocortex, this surface can be 'inflated' or remapped, for example, for viewing on the surface of a sphere. Registration can then be performed between two spheres based on some feature map by rotating until the maximum overlap achieved (*Kim et al., 2005*; *Klein et al., 2010*; *Lyttelton et al., 2007*). Posing registration problems on a sphere helps account for inter-individual variability in gyral and sulcal patterning, which can vary drastically between individuals (*Bartley et al., 1997*; *Le Goualher et al., 1999*; *Régis et al., 2005*). For example, conventional 3D volumetric registration may take one gyrus and stretch it across two gyri from another individual, especially in areas where the number or shape of gyri varies between individuals. In surface-based registration, homologous points are not constrained to fall in similar absolute positions but rather similar topological positions (e.g., one gyral peak could be homologous to a point halfway down the depth of an adjacent sulcus from another individual). Since the major gyri and sulci of the brain are typically invariant across individuals (*Le Guen et al., 2018*), gyral and sulcal patterning can be used as one feature to inform registration, often after smoothing to remove smaller secondary or tertiary gyri and sulci which tend to be more variable (*Tardif et al., 2015*), while other features more indicative of cortical architecture, such as thickness or intracortical myelin, can be used as well (*Glasser et al., 2016*; *Lyttelton et al., 2007*; *Van Essen et al., 2012*).

Here, we present such a surface-based registration method specifically for hippocampal surfaces. Rather than inflation to a sphere, we rely on previous work which maps the hippocampus to a flat rectangle to preserve its topology (*DeKraker et al., 2022*). This also allows for the use of existing 2D image-based registration tools without reformulation for use on a surface mesh but, in effect, the same advantages and constraints as typical surface-based registration are preserved. Evaluation of this method is performed using ground-truth (i.e., histologically derived) subfield segmentations from seven samples that were sliced, imaged microscopically, and then digitally reconstructed into a 3D block with various histology contrasts. We benchmark this new method against unfolding alone, which provides some intrinsic alignment between subjects (*DeKraker et al., 2018*) but which we believe can be further improved with the present methods, and against more conventional 3D volumetric registration approaches. The method has been openly published at https://github.com/khanlab/hippunfold/releases/tag/v1.3.0 (*DeKraker and Khan, 2023*).

## Materials and methods

### Data

Seven 3D reconstructed hippocampal histology samples were examined in this study from three different datasets, including four donor brains:

1. BigBrain: two hemispheres (donor 1), Merker stain (*Merker, 1983*), downsampled from 20 to 40 µm isotropic voxel resolution (*Amunts et al., 2013*).
2. 3D polarized light imaging (3D-PLI): one hemisphere (donor 2), 48 × 48 × 60 µm resolution, contrast driven by birefringence properties of myelin sheaths surrounding axons (*Axer et al., 2011*).

3. AHEAD: four hemispheres (donors 3 and 4), blockface imaging and multiple stains, 150 × 150 × 200 μm resolution (*Alkemade et al., 2022*).

Details of these dataset acquisitions and processing can be found in their respective references. The 3D-PLI sample was examined here only in terms of transmittance rather than directional data. The transmittance has already been demonstrated in a few brain sections to provide valuable information to segment hippocampal subregions (*Zeineh et al., 2017*).

## Manual segmentation

HippUnfold (*DeKraker et al., 2022*) achieves automatic segmentation and unfolding of in vivo hippocampal MRI data, typically T1w or T2w images, which do not resemble the contrasts seen in the present histology datasets and thus manual segmentation had to be used. This unfolding method requires a detailed hippocampal gray matter mask, as well as labeling of termini at the anterior, posterior, and medial hippocampal edges, dentate gyrus (DG), and laminar strata radiatum and lacunosum-moleculare (SRLM). Here, we consider SRLM to be a 'mixed' label since it can include components of the subicular complex, CA fields, DG, as well as blood vessels and CSF within the hippocampal sulcus. Thus, it is used to differentiate the upper and lower surfaces of the remaining hippocampal cortex, contiguous with the 'pial' and 'white' surfaces of the neocortex, respectively, and SRLM volume is excluded from further analyses.

In BigBrain, these labels were available from previous work (*DeKraker et al., 2020*). Rater J.D. performed manual segmentation in all other samples using ITK-SNAP (*Yushkevich et al., 2006*). In

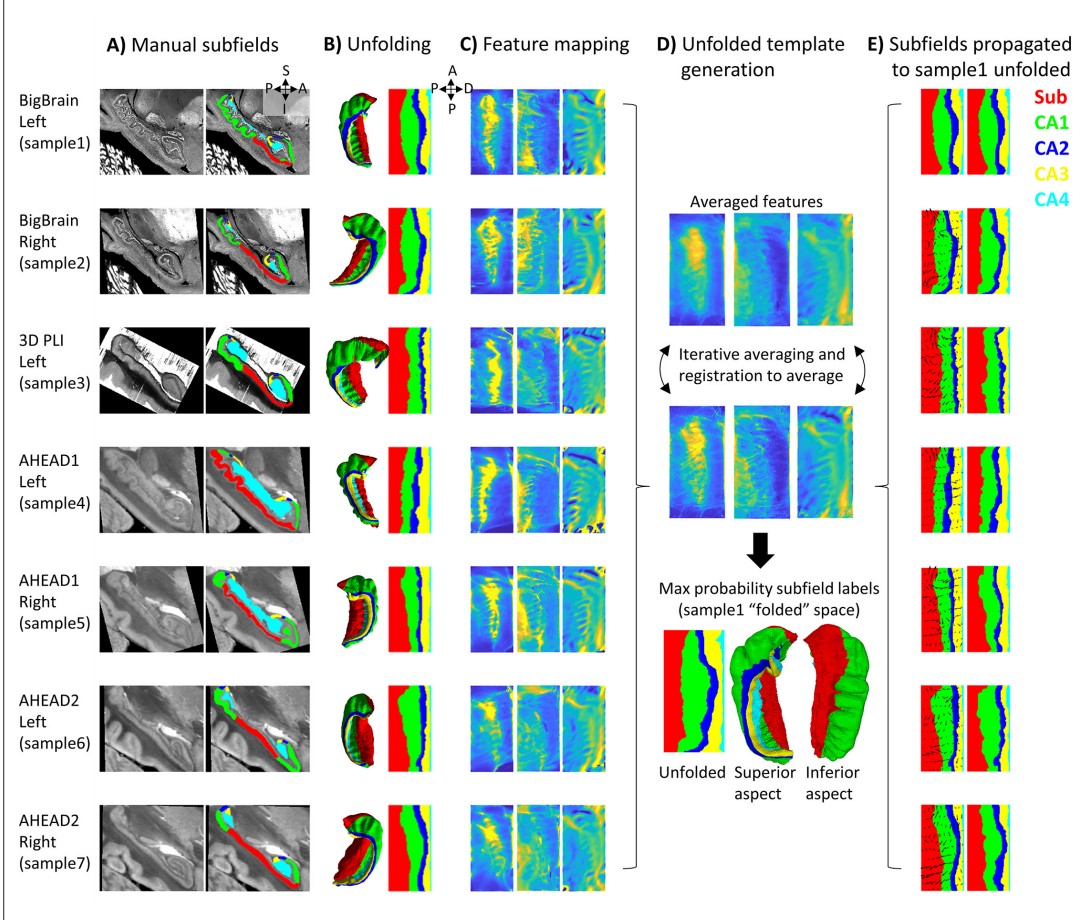

**Figure 1.** Surface-based subfield alignment pipeline. (**A**) Each 3D dataset manually parcellated into subfields. (**B**) Subfields mapped to a common unfolded space using HippUnfold. (**C**) Morphometric features mapped to unfolded space (from left to right: gyrification, thickness, curvature). (**D**) Multimodal features iteratively aligned in unfolded space using 2D registration. (**E**) Subfield labels propagated through unfolded 2D registrations to sample1. Closeups of the segmentations for the individual datasets can be found in Appendix 2.

the 3D-PLI sample, the absolute anterior and posterior of the hippocampus were cut off during tissue preparation. Thus, the required labels were extrapolated manually over the missing regions in order to recover a fully 3D hippocampal shape that is amenable to unfolding (see *Figure 1A*). In the AHEAD brain samples, manual segmentation was performed on blockface images.

For all datasets, hippocampal gray matter was subsequently manually parcellated by rater J.D. into subfield labels: subicular complex (Sub) and cornu ammonis (CA) fields 1–4 based on the available histological features and according to the criteria outlined by *Ding and Van Hoesen, 2015*; *Duvernoy et al., 2013*; *Palomero-Gallagher et al., 2020*. Since these sources differ slightly in their boundary criteria, and no prior reference perfectly matches the present samples, subjective judgment was used to draw boundaries after considering all three prior works. The 'prosubiculum' label used by Ding and Van Hoesen and Palomero-Gallagher et al. was included as part of the subicular complex. See Appendix 2 'Ground-truth segmentation' for more details. The DG was also labeled but was grouped together with CA4 since HippUnfold's current method for unfolding of the DG relies on heuristics in MRI that would not be appropriate in this work (i.e., it uses a template prior to estimate DG topology at the cost of smoothing labels). These manual labels were defined based upon cytoarchitectonic features at the highest level of resolution available and were deemed 'ground-truth' subfield definitions. It is important to note that BigBrain is stained for cell bodies, while 3D-PLI transmittance contrast is driven by cell bodies and nerve fibers (both introducing light extinction effects; *Menzel et al., 2020*) and thus contains very different microstructural information. Thus, both cell body and fiber distribution patterns were consulted during subfield definition. In the AHEAD dataset, multiple imaging modalities were available, albeit with imperfect registration to the blockface images, larger interslice gaps, some missing data, and limited resolution. These additional contrasts were overlaid over blockface images (where available and appropriate due to the above limitations) to better inform subfield segmentation.

## Unfolded registration

HippUnfold (*DeKraker et al., 2022*) was used to map each dataset to a standardized unfolded space (see *Figure 1B*). This unfolded space consists of a triangulated mesh with uneven face sizes, so as to preserve a constant spacing between points in the folded hippocampus. The current work, however, defined this tessellation as a regular mesh grid in unfolded space consisting of 256 × 128 points across the anterior-posterior (A-P) and proximal-distal (P-D) (relative to the neocortex) axes of the unfolded hippocampus, respectively. This regular grid in unfolded space means that surface points (henceforth, vertices) can effectively be treated as pixels of a flat 2D image without the need to interpolate missing pixel values. However, it should also be noted that, as a consequence, native or 'folded' images are sampled more densely in some areas than others, particularly in the anterior and posterior extremes, which can lead to noisier unfolded data in these regions.

HippUnfold also calculates morphological features, namely thickness, gyrification index, and curvature in each subject's native space (*Figure 1C*). These features are desirable for inter-subject registration since they (i) are associated with subfield boundaries (*DeKraker et al., 2018*; *DeKraker et al., 2022*; *Yushkevich et al., 2015b*), (ii) do not require cytoarchitectonic information to measure, and (iii) are agnostic to imaging contrast differences. Registration performed in unfolded hippocampal space is analogous to registration of neocortical surfaces that have been inflated to a sphere since both methods preserve topology rather than absolute position. However, one difference is that registration on a sphere allows one point to 'wrap' around the meridian of a sphere whereas in a rectangular unfolded space, the proximal edge (i.e., closest to the neocortex) does not 'wrap' to the distal (i.e., closest to the dentate gyrus) edge of the P-D axis, and the same applies to the A-P edges. This is in agreement with the true geometry of the hippocampus though, which has the topology of a rectangle (i.e., four true edge termini) rather than a sphere (i.e., zero true termini).

The 2D registration was performed using all three of the above morphological features with equal weighting, using ANTs multicontrast SyN deformable registration (*Avants et al., 2011*; *Figure 1D*). Rather than registering all samples' feature maps to one sample, we instead used an iterative template building method (*Avants et al., 2010*), which first averages images, registers each image to the average, and then repeats the registration to the newly generated average. This process is repeated four times, with each iteration sharpening the averaged template and improving registration precision. We concatenated the transforms from each sample to the template with the inverse transform

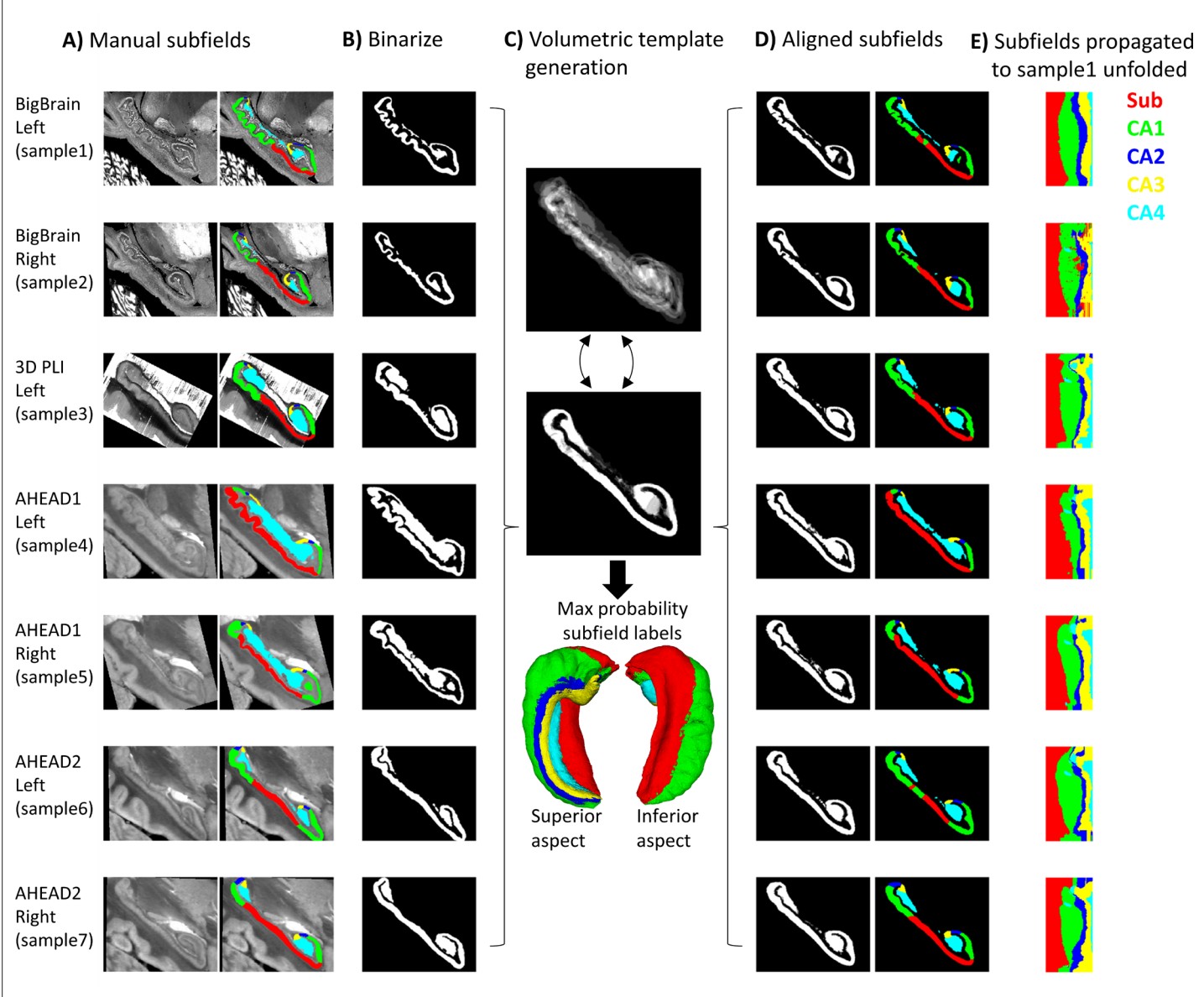

**Figure 2.** Control condition using volumetric registration. (**A**) Subfield segmentation as in **Figure 1A**. (**B**) Binarized and rigidly aligned hippocampal gray matter masks (left hemispheres flipped). (**C**) Iterative alignment using ANTs template building in 3D. (**D**) Each sample's subfields propagated to template space. (**E**) Each sample's subfields propagated to sample1 unfolded space.

from the template to sample1 (BigBrain left hemisphere) and applied it to subfield labels to evaluate their overlap with that sample's ground-truth subfield definitions (**Figure 1E**). Sample1 was chosen because it had the highest resolution and, therefore, provided the greatest cytoarchitectonic detail for identification of subfield boundaries, while also being the more common hemisphere in this study (i.e., four left; three right). In principle, any sample could have been chosen, and one had to be chosen to test overlap in any one native (or folded) space.

## Control condition: Volumetric registration

The proposed pipeline was compared to a conventional 3D volumetric registration approach: ANTs template generation (**Avants et al., 2011**) under ideal tissue contrast conditions (i.e., based on binarized gray matter labels). This is outlined in **Figure 2** and detailed below. First, all hippocampal gray matter labels were binarized, right hippocampi were flipped, and binary masks were rigidly registered to sample1 using Greedy's moment-based initialization (**Yushkevich et al., 2016**) (two moments),

which can handle images initially in different spaces relative to the origin (which was the case in some samples here) (*Figure 2B*). Images were then resampled to 100 µm isovoxel resolution to reduce compute time and memory requirements, which were prohibitively high at native resolution. ANTs template generation was then used as before (*Figure 2C*), with the following differences: registrations were all 3D and unimodal.

Using binarized images likely overestimates a fully automated registration method for subfield parcellation, since it presents idealized tissue contrast conditions (i.e., perfect contrast between gray and white matter). This has been chosen as a practical and robust approach, avoiding local minima commonly encountered by image-intensity based registration procedures in the mesiotemporal region (*Qiu and Miller, 2008*). In principle, cross-modal registration can be performed using metrics like mutual information or cross-correlation, but these metrics still do not fully compensate for differences in contrasts, luminance, or intensity levels, and may have many local minima solutions making it less tractable. Thus, the control condition used here could be thought of as a best-case volumetric registration.

Registrations from each sample to the template and the inverse transform from the template to sample1 were concatenated and applied to subfield labels (*Figure 2D*). For easier comparison with *Figure 1*, each sample's subfields were sampled along sample1 mid-thickness surface and projected to unfolded space (*Figure 2E*). In some cases, the mid-thickness surface fell outside of the propagated subfield labels and returned a background value of 0. The missing values were imputed by nearest-neighbor interpolation in unfolded space, providing additional correction for misregistration.

## Evaluation metrics

The Dice overlap metric (*Dice, 1945*), which can also be considered an overlap fraction ranging from 0 to 1, was calculated for all subjects' subfields registered to sample1. This was repeated in 2D unfolded and in 3D native spaces since some parts of unfolded space expanded or contracted more than others when projected to native space, which can over- or under-emphasize subfield differences. For example, sample3 was unfolded and then registered to the unfolded average, making up two transformations. These were then concatenated with the inverse transformation of unfolded sample1 to the same unfolded average, and the inverse transformation of native sample1 to unfolded space. This concatenated transformation was used to project labels from sample3 native space directly to sample1 native space, which should ideally lead to near-perfect subfield alignment in sample1 native space. Dice overlap between sample1 and sample3 registered to sample1 was then calculated in sample1 native space.

A secondary metric, border distances, was also calculated in sample1 native space. This was calculated by computing distance from a given border in sample1 to all voxels, followed by concatenating these distances at the location of each propagated sample's corresponding subfield borders, providing a minimum direct 3D distance between borders in real-world units. These distances were calculated from all subfield borders (i.e., Sub-CA1, CA1-CA2, CA2-CA3, and CA3-CA4).

In addition to registration in 2D unfolded space and 3D native space, overlap metrics were calculated for unfolding without registration. That is, borders from each sample were projected to unfolded space and projected directly to sample1 space, which replicated the current subfield segmentation behavior in HippUnfold (v1.2.0).

To evaluate the contribution of each morphometric feature or combination of features (thickness, gyrification, and curvature) to registration, unfolded space registrations were also performed using all combinations of these features. These were then evaluated using Dice scores in sample1 native space.

## Results
### Qualitative alignment

*Figure 1A* shows equivalent sagittal slices from each sample after rigid alignment, in which considerable subfield variability can be seen between samples. This is due in part to the out-of-plane issue discussed elsewhere (*DeKraker et al., 2021*). Viewing a fully 3D model (*Figure 1B*) can help in identifying differences between samples due to true morphological variability rather than field-of-view differences. Projecting these labels to unfolded space (*Figure 1B*) preserved sample-specific differences in the size of each subfield but removed variability due to different folding and gyrification

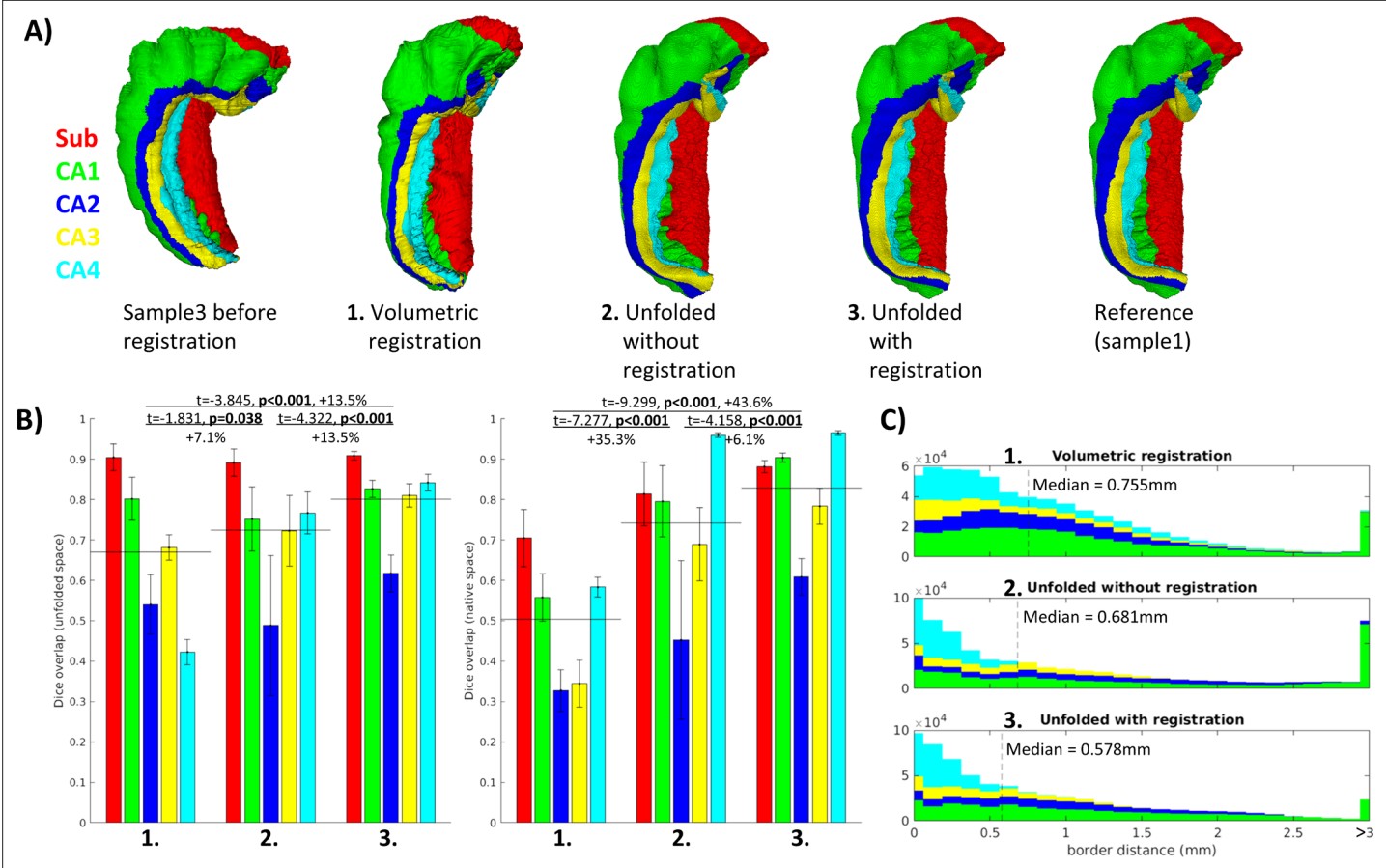

**Figure 3.** Evaluation of aligned subfield definitions. (**A**) Qualitative example of subfields from the third sample projected to the first sample's native 'folded' space using conventional 3D volumetric alignment, unfolding to account for inter-individual differences in folding shape, and unfolding followed by registration in unfolded space. (**B**) Dice overlap achieved. Each measure was calculated in unfolded space (left) and again in the first sample's (BigBrain left hemisphere) native folded space (right). Black lines indicate the mean across all subfields. (**C**) Distances between all aligned subfield borders using the three methods described above. Dashed lines indicate the median distance.

configurations between samples, which already brought subfields into close alignment and was the current basis for subfield segmentation in HippUnfold.

Following registration in unfolded space (*Figure 1E*), subfields are even more closely aligned (e.g., samples 3 and 5 are no longer dominated by the Sub label). In addition, there are no topological breaks or reordering of subfields in *Figure 1*. In *Figure 2E*, following conventional volumetric alignment, there are cases where CA1 shows isolated islands (e.g., sample2), and where CA1 borders CA3 directly instead of first passing through CA2 (e.g., sample3). This does not match the original subfield segmentations from any sample or the literature which states that subfields should be contiguous and consistently ordered (*Duvernoy et al., 2013*). These issues arise in 3D registration due to breaks in topology: for example, hippocampal gray matter may become stretched across the SRLM and vestigial hippocampal sulcus, or across adjacent gyri. When subfield labels are propagated across a sulcus, they can become discontinuous with respect to the mid-thickness surface topology of a given sample. This type of 2D topology is conserved in a surface-based or unfolded hippocampal space registration.

## Quantitative alignment

Better Dice overlap on average and for every individual subfield was observed using unfolded registration over unfolding and refolding into a different sample's folded configuration alone (*Figure 3B*). Both methods also outperformed the control condition using conventional ANTs 3D volumetric registration. These results were tested using one-tailed, paired-samples *t*-tests, pairing subfields and subjects, which revealed significant differences between each method and in both unfolded and native

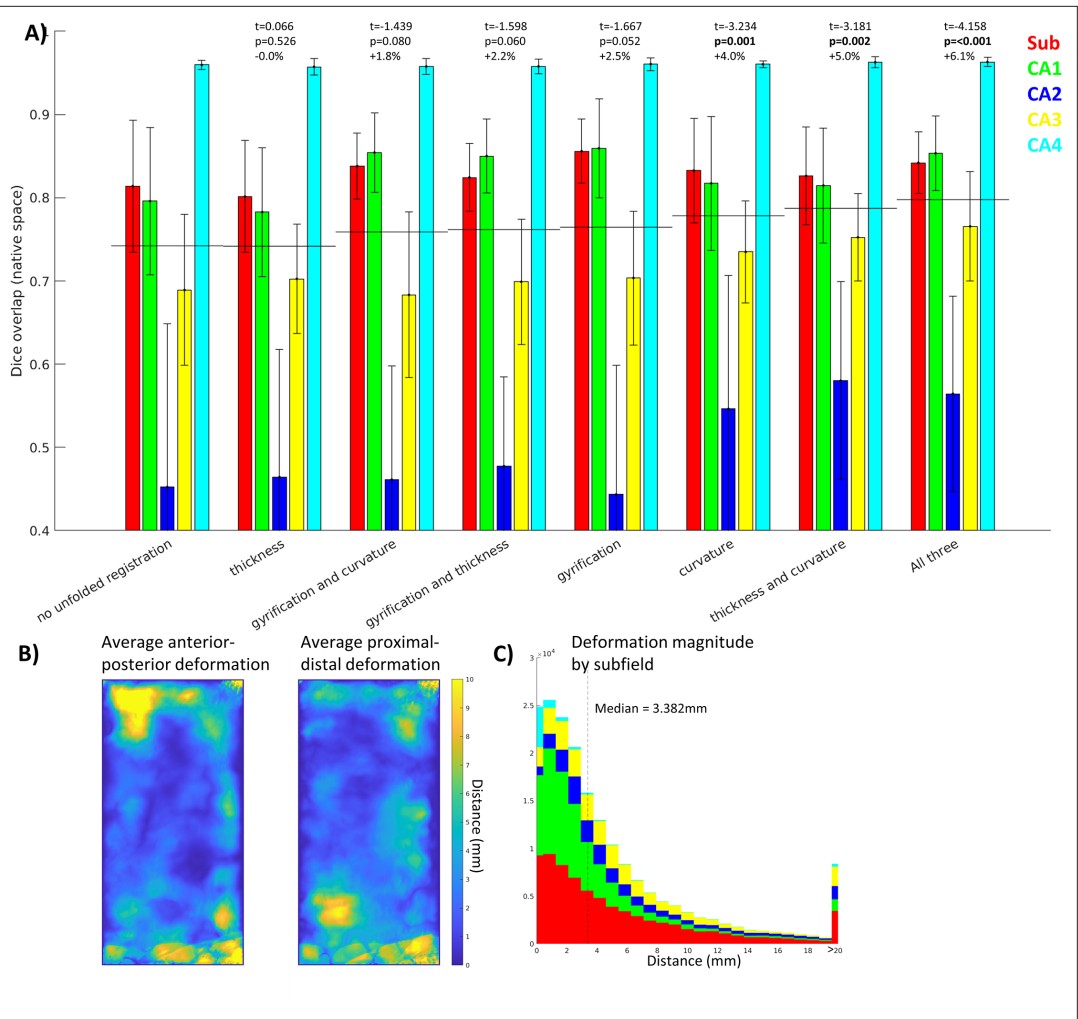

**Figure 4.** Contribution of each morphological feature to unfolded registration performance. (**A**) Unfolded space registration was repeated for all combinations of unfolded morphological features and evaluated by Dice overlap in native space. Combinations are ordered by their Dice scores averaged across the five subfields. p-values are relative to no unfolded registration, using one-tailed paired-samples t-tests as above. (**B**, **C**) Evaluation of which hippocampal vertices (**B**) and subfields (**C**) were most deformed in unfolded registration.

spaces. The same pattern was also observed when evaluated according to nearest corresponding subfield border distances in native space (*Figure 3C*). One example of each of these registration methods (sample3 to sample1) is shown in *Figure 1A*.

## Contribution of unfolded morphological features

By using different combinations of unfolded features, we could determine which is most informative about unfolded registration and subfield boundary alignment (*Figure 4*). In this regard, curvature was the most informative individual feature while thickness and curvature was the most informative combination of two features, despite the fact that thickness was the least informative individual feature (similar to no unfolded registration). Therefore, while each is informative, they may contain overlapping information and their combination is not as complementary as curvature and thickness together. Regardless, combining all three features still showed the best performance. Future work should explore the use of additional features, such as intracortical myelin or laminar distributions of neurons, to inform registration. This was not examined here since it was not available in all datasets owing to differing contrasts.

*Figure 4B and C* show the extent of deformations in unfolded space following ANTs registration with default parameters. The median deformation magnitude across all vertices was 3.382 mm, but

more constrained or liberal deformations could be achieved by adjusting the ANTs elasticity and fluidity parameters (proportional to the resulting deformation field's maximum and smoothness, respectively). More liberal deformations could lead to more precise subfield alignment; however, this also runs the risk of falling into local minima during optimization or of distorting tissue beyond a reasonable distance. Thus, default parameters were used to maintain robustness at the cost of potential gains in precision.

## Discussion

The current work presented a novel surface-based registration of hippocampal cortex between histology samples, and our evaluations demonstrated that it outperformed even an idealized version of conventional volumetric registration. This enabled mapping of multiple features at a sub-millimetric scale that would otherwise be impossible. For example, maps of hippocampal cyto- or myeloarchitectonic features can be constructed from multiple samples using different stains or imaging methods that would otherwise preclude one another due to tissue destruction (e.g., once stained, a slice cannot be easily imaged with other stains). This can be useful both in studying hippocampal subfields and in subfield-agnostic mapping and other data driven methods (e.g., *Borne et al., 2023*; *Paquola et al., 2020*; *Patel et al., 2020*; *Przeździk et al., 2019*; *Vogel et al., 2020*; *Vos de Wael et al., 2018*). We hope that this work will provide an avenue toward mapping of hippocampal data across many modalities, scales, and different fields in future work.

Surface-based hippocampal registration can be used in subfield parcellation in MRI, histology, or other imaging methods by registration to the unfolded maximum probability subfield atlas provided here (*Figure 1D*). Note that the current work differs from other subfield segmentation protocols, even those which employ surfaces and unfolding, in that our method constrains registration (and therefore subfield segmentation) topologically by the unique folded shape of a given hippocampus. Other methods generally first employ subfield segmentation (either manually or using conventional volumetric registration) and then reduce volumes to surfaces or employ other flat mapping techniques of the hippocampus (*Caldairou et al., 2016*; *Ekstrom et al., 2009*; *Pipitone et al., 2014*; *Yushkevich et al., 2016*; *Yushkevich et al., 2015b*; *Zeineh et al., 2000*; *Zeineh et al., 2001*). For example, Surf-Patch (*Caldairou et al., 2016*) computes volumetric registration and then propagates surfaces rather than labelmaps, which avoids discretization errors. However, this method still does not guarantee correct topology across different hippocampal folding patterns (see discussion below on multi-atlas registration) and has not been demonstrated at the level of detail examined here.

Other surface-based registration methods such as spherical harmonics (SPHARM) (*Brechbühler et al., 1995*; *Gerig et al., 2001*) have been employed to find homologous vertices between irregular shapes including hippocampal samples (*Styner et al., 2004*). In principle, this is a similar pose of the registration problem as the methods employed here. However, SPHARM requires a spherical topology, which in *Styner et al., 2004* is mapped to the outer boundaries of the hippocampus rather than a midthickness surface, and so this method does not fully leverage the geometric topology constraints of the hippocampus. There has been an adaption of the SPHARM-PDM model to hippocampi, in which the spherical parameterization of the outer hull was propagated along a Laplacian field to the hippocampal midthickness surface (*Kim et al., 2014*), and this approach has since then been used and validated in the study of hippocampal organization in both health and disease (*Bernhardt et al., 2016*; *Vos de Wael et al., 2018*). Other vertex-wise or even point-cloud registration methods could be employed for hippocampal midthickness surfaces in future work. One final example is the recent FastSurfer implementation of Laplace Eigenfunctions for neocortical surface registration, which involves registration to a sphere (*Henschel et al., 2020*). This method does not require an inherently spherical topology and the only major conceptual difference between it and the present work is that we hold hippocampal termini or endpoints fixed, for additional regularization, whereas FastSurfer derives them from the surface mesh itself.

*Ravikumar et al., 2021* recently performed flat mapping of the medial temporal lobe neocortex using a Laplace coordinate system as employed here and showed sharpening of group-averaged images following deformable registration in unfolded space. This indirectly suggests better inter-subject alignment. We perform a similar group-averaged sharpening analysis in Appendix 1 'In vivo demonstration.' Though the gains in image sharpness observed here were modest, we note that current MRI resolution and automated segmentation methods allow for only imperfect hippocampal

feature measures. We thus expect unfolded registration to grow in importance as MRI and segmentation methods continue to advance.

The feature most strongly driving surface-based registration in the present study was curvature. Many neocortical surface-based registration methods focus on gyral and sulcal patterning at various levels (e.g., strong alignment of primary sulci, with weaker weighting on secondary and tertiary sulci) (*Miller et al., 2021*). In the present study, hippocampal gyri are variable between samples and so could perhaps be thought of as similar to tertiary neocortical gyri, and this may help explain why gyrification was not the primary driving feature in aligning hippocampal subfields. The methods used to quantify gyrification are often related to curvature, but differ across studies. In the hippocampus, unlike in the neocortex, the mouth of sulci are wide and so sulcal depth, which is often used in defining neocortical gyrification index, is not straightforward to measure. Using HippUnfold, gyrification is defined by the extent of tissue distortion between folded and unfolded space, and individual gyri/sulci are hard to resolve in unfolded gyrification maps, but are readily visible in curvature maps. Thus, hippocampal curvature may be more informative simply due to higher measurement precision. Future work could also employ measures like quantitative T1 relaxometry or other proxies of intracortical myelin content (e.g., *Tardif et al., 2015*; *Glasser et al., 2016*; *Paquola et al., 2019*) for hippocampal alignment, but this is not possible in cross-modal work as in the various histology stains examined here.

One limitation of the evaluation performed on our surface-based registration is that our control condition using 3D volumetric registration did not employ a multi-atlas as in some other popular methods, including those discussed above (*Caldairou et al., 2016*; *Yushkevich et al., 2016*; *Yushkevich et al., 2015b*). A multi-atlas applies registration of several references to a target image and then combines propagated labels from the references to the target (e.g., via maximum probability). We partially compensated for this issue by using an idealized volumetric registration with detailed binarized hippocampal gray matter masks rather than images, which generally contain more noise, blurring, and surrounding structures that are not necessarily informative about hippocampal shape. 3D registrations at the current resolution also have costly compute requirements that scale with resolution and the number of samples in the multi-atlas. In addition, a multi-atlas is not guaranteed to contain a sample with a similar folding configuration as the target sample and, even if it does, the combination of multiple registered samples may lead to errors or over-smoothing. By contrast, HippUnfold can be moderately compute intensive at high resolution, but it only needs to be performed once and then registration in unfolded space has trivially light compute requirements, even when using multiple contrasts as performed here. Nevertheless, we hope to eventually see the development of multi-atlas volumetric registration at a microscale, as well as work performing a systematic comparison with surface-based registration.

In conclusion, we formulated a registration performed in a standardized 'unfolded' hippocampal space and showed that this method consistently improved inter-individual alignment with respect to subfields. This method is topologically constrained and driven by contrast-agnostic feature maps, meaning that it can be performed across image modalities regardless of whether cytoarchitectonic features are directly accessible or not. Overall, this work constitutes a state-of-the-art registration method between hippocampi at a scale approaching the micron level.

## Acknowledgements

This work was supported by the Helmholtz International BigBrain Analytics and Learning Laboratory (HIBALL), funded by Healthy Brains, Healthy Lives (HBHL) and the Helmholtz Association. JD was furthermore supported by the National Science and Engineering Research Council of Canada (NSERC). BCB acknowledges research support from NSERC (Discovery-1304413), Canadian Institutes for Health Research (CIHR FDN-154298, PJT-174995), SickKids Foundation (NI17-039), Brain-Canada, HBHL, and the Tier-2 Canada Research Chairs program. 3D-PLI-related work was supported by the European Union's Horizon 2020 Framework Programme for Research and Innovation (grant no. 945539): 'Human Brain Project' SGA3 and by the computing time granted through JARA on the supercomputer JURECA at Forschungszentrum Jülich.

# Additional information

## Funding

| Funder | Grant reference number | Author |
| --- | --- | --- |
| Natural Sciences and Engineering Research Council of Canada | Post doctoral fellowship | Jordan DeKraker |
| Natural Sciences and Engineering Research Council of Canada | Discovery-1304413 | Boris C Bernhardt |
| Canadian HIV Trials Network, Canadian Institutes of Health Research | FDN-154298 | Boris C Bernhardt |
| Sickkids Research Institute | NI17-039 | Boris C Bernhardt |
| Horizon 2020 Framework Programme | 945539 | Markus Axer |
| Canadian HIV Trials Network, Canadian Institutes of Health Research | PJT-174995 | Boris C Bernhardt |

The funders had no role in study design, data collection and interpretation, or the decision to submit the work for publication.

## Author contributions

Jordan DeKraker, Conceptualization, Resources, Data curation, Software, Formal analysis, Funding acquisition, Investigation, Visualization, Methodology, Writing – original draft, Writing – review and editing; Nicola Palomero-Gallagher, Data curation, Validation, Investigation, Methodology, Writing – review and editing; Olga Kedo, Resources, Data curation, Validation; Neda Ladbon-Bernasconi, Conceptualization, Validation; Sascha EA Muenzing, Resources, Data curation; Markus Axer, Resources, Data curation, Methodology; Katrin Amunts, Conceptualization, Validation, Writing – review and editing; Ali R Khan, Software, Methodology; Boris C Bernhardt, Conceptualization, Formal analysis, Supervision, Funding acquisition, Visualization, Writing – original draft, Project administration, Writing – review and editing; Alan C Evans, Conceptualization, Formal analysis, Supervision, Funding acquisition, Investigation, Writing – original draft, Project administration, Writing – review and editing

## Author ORCIDs

Jordan DeKraker http://orcid.org/0000-0002-4093-0582
Nicola Palomero-Gallagher http://orcid.org/0000-0003-4463-8578
Katrin Amunts http://orcid.org/0000-0001-5828-0867
Ali R Khan http://orcid.org/0000-0002-0760-8647
Boris C Bernhardt https://orcid.org/0000-0001-9256-6041

## Ethics

This work was conducted using opensource human data from various institutions, including consent to share data openly, for which Research Ethics Board permissions were obtained though each institution, respectively.

Reviewer #1 (Public Review): https://doi.org/10.7554/eLife.88404.4.sa1
Reviewer #2 (Public Review): https://doi.org/10.7554/eLife.88404.4.sa2
Author Response https://doi.org/10.7554/eLife.88404.4.sa3

# Additional files

## Supplementary files

• MDAR checklist

• Supplementary file 1. Side-by-side comparison between raters (O.K. top; J.D. middle; unlabeled bottom) in a subset of 54 slices where both the left and right hemispheres were fully labeled. Slices are ordered from anterior-to-posterior.

## Data availability

All code and data for the analyses performed in this study are openly available at https://zenodo.org/record/7757416. The opensource tool HippUnfold has been updated to employ the methods developed in this study. This tool can be found at https://github.com/khanlab/hippunfold (copy archived at *Khan Computational Imaging Lab, 2023*).

The following dataset was generated:

| Author(s) | Year | Dataset title | Dataset URL | Database and Identifier |
|---|---|---|---|---|
| DeKraker J, Palmero-Gallagher N, Kedo O, Ladbon-Bernasconi N, Muenzing SEA, Axer M, Khan AR, Bernhardt B, Evans AC | 2023 | Evaluation of surface-based hippocampal registration using ground-truth subfield definitions | https://doi.org/10.5281/zenodo.7757415 | Zenodo, 10.5281/zenodo.7757415 |

The following previously published datasets were used:

| Author(s) | Year | Dataset title | Dataset URL | Database and Identifier |
|---|---|---|---|---|
| Amunts K, Lepage C, Borgeat L, Mohlberg H, Dickscheid T, Rousseau MÉ, Bludau S, Bazin PL, Lewis LB, Oros-Peusquens AM, Shah NJ | 2013 | BigBrain: an ultrahigh-resolution 3D human brain model | https://bigbrain.loris.ca/main.php | LORIS, bigbrain |
| Bazin PLEA, Forstmann BU, Alkemade A | 2022 | Ahead brain 122017 - 3D reconstructions | https://doi.org/10.21942/uva.16834114.v2 | University of Amsterdam / Amsterdam University of Applied Sciences, 10.21942/uva.16834114.v2 |
| Bazin PLEA, Forstmann BU, Alkemade A | 2022 | Ahead Brain 152017 - 3D reconstructions | https://doi.org/10.21942/uva.14260064.v2 | University of Amsterdam / Amsterdam University of Applied Sciences, 10.21942/uva.14260064.v2 |

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

# Appendix 1

## In vivo MRI demonstration

In order to determine whether the gains from unfolded registration introduced in the current histology work translate to in vivo MRI, we examined a set of 10 healthy participants scanned at 7 Tesla magnetic field strength. Each subject's T1w image was run through the automated HippUnfold pipeline (*DeKraker et al., 2022*) with and without the inclusion of unfolded space registration to the group-averaged template derived in the present study. We reasoned that including unfolded registration should lead to better alignment and therefore sharper images after averaging across subjects and hemispheres, as demonstrated, for example, by *Ravikumar et al., 2021*. Here, we quantify sharpness as the mean gradient magnitude (MGM) of the averaged image. Morphological features used in this unfolded space alignment, namely, gyrification, thickness, and curvature, showed a 10.9, 0.6, and 9.1% increase in MGM, respectively.

We also measured quantitative T1 relaxation times (qT1) along hippocampal midthickness surfaces. This generally showed differences across the proximal-distal, or subfield-related, axis of the hippocampus, with highest values being found in the CA1 region indicating a relatively low intracortical myelin content compared to the other subfields. Group-averaged qT1 MGM also increased, by 2.1%, with the inclusion of registration in unfolded space. This feature was not used to inform unfolded registration but still showed a small increase in sharpening with group averaging, which reinforces the validity of morphometry as a basis for registration across many domains.

It should be noted that the gains from unfolded registration are likely curtailed in in vivo MRI compared to in histology due to decreased resolution and therefore reduced sensitivity of morphological measures. Even at 7 Tesla MRI field strength, it was noted that considerably less detail (namely, less small folds or gyrifications within the hippocampus) were visible both in the raw images and in the automatically generated image segmentations and surfaces. This leads to overall lower gyrification and curvature, and increased thickness measures. We thus note that the gains in intersubject alignment demonstrated here may not be as clear in more commonly used 3 Tesla MRI, but we hope that as acquisition methods continue to advance, the alignment methods demonstrated here will only grow in importance.

## Methods

All participants were recruited between May 2022 and April 2023 at the Montreal Neurological Institute (MNI). Healthy controls met the following inclusion criteria: (1) age between 18 and 65 y; (2) no neurological or psychiatric illness; (3) no MRI contraindication; (4) no drug/alcohol abuse problem; and (5) no history of brain injury and surgery. The Research Ethics Board (REB) at McGill University approved this study. Ten participants were examined here (mean age 26.7 y, S.D. 4.36).

Scans were acquired using a 7 Tesla Siemens MRI at the McConnell Brain Imaging Centre of the Montreal Neurological Institute. Each participant underwent multiple types of scans, including T1-weighted (T1w) structural MRI, diffusion-weighted imaging (DWI), resting-state functional MRI (rs-fMRI), and quantitative T1 (qT1) mapping. For the purposes of the present study, we will discuss only qT1 imaging.

qT1 relaxometry data was acquired using a 3D-MP2RAGE sequence, with the following parameters: 0.5 mm isotropic voxels, 320 sagittal slices, TR = 5170 ms, TE = 2.44 ms, TI1 = 1000 ms, TI2 = 3200 ms, flip angle = 4°, flip angle 2 = 4°, iPAT = 3, bandwidth = 210 Hz/px, echo spacing = 7.8 ms, and partial Fourier = 6/8. Both inversion images were combined for qT1 mapping to minimize sensitivity to B1 inhomogeneities and optimize intra- and inter-subject reliability (*Haast et al., 2016*; *Marques et al., 2010*).

HippUnfold was run on raw qT1 images. The inclusion of unfolded space registration was performed using the latest HippUnfold software release (v1.3.0, described fully at https://github.com/khanlab/hippunfold/releases/tag/v1.3.0; *DeKraker and Khan, 2023*) which includes the methods and reference histology data described in the current study.

Outliers (3 S.D. from the median) were clipped for each feature before averaging. Each group-averaged feature (gyrification, thickness, curvature, and qT1) was z-scored before calculating the image gradient and MGM. This makes the MGM ranges more similar for easier comparison across the different features.

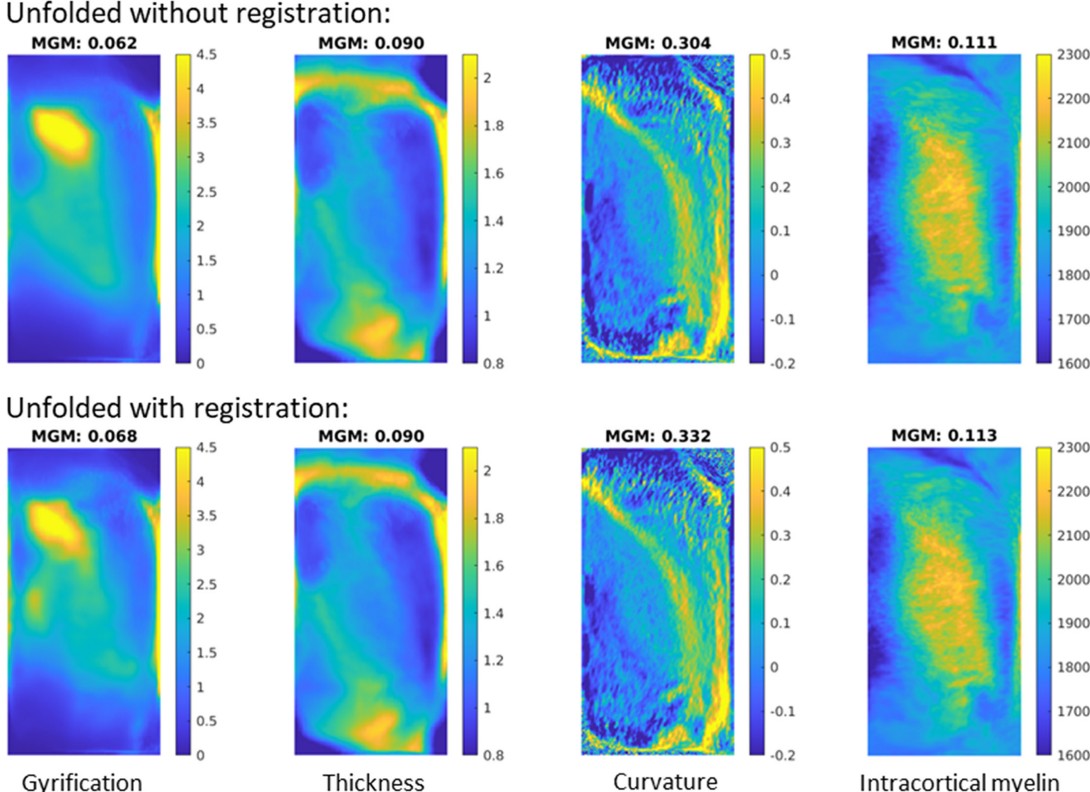

**Appendix 1—figure 1.** Image sharpness when averaging in vivo MRI subjects' hippocampal features with and without unfolded space registration. MGM = mean gradient magnitude of the group-averaged image.

# Appendix 2

## Ground-truth segmentation

As discussed in the 'Introduction' section, the definition of hippocampal subfields in MRI differs systematically between labs (see *Yushkevich et al., 2015a*), and this is also the case in 'ground-truth' histological subfield definitions (*Olsen et al., 2019*). To our knowledge, these differences have not been quantified systematically, but *Olsen et al., 2019* demonstrate considerable differences between raters even with a coronal slice where no out-of-plane sampling issues are present. The evaluation of unfolded hippocampal subfield registration detailed in the present study does not rely on harmonized subfield definitions, but rather on internal (or intra-rater) consistency and were thus all performed by J.D. However, we nevertheless sought to determine whether these definitions matched those of other neuroanatomists in the field in order to determine the value of the current ground-truth labelmaps as a reference material for future work. We have made labelmaps for all samples in native and unfolded spaces and the maximum probability unfolded subfield labels available online (see 'Data availability statement').

Expert histologist O.K. performed manual subfield annotations in a subset of 66 coronal slices from the BigBrain left and right hemispheres. O.K.'s annotations included labels 'parasubiculum,' 'subiculum proper,' 'presubiculum,' and 'prosubiculum' *Palomero-Gallagher et al., 2020*; *Amunts et al., 2021*; doi: 10.25493/X4S6-E64 which were all combined into a single label to match the 'subicular complex' label employed by rater J.D. Additionally, to focus on borders between subfields rather than differences in gray matter definition, the same mask of the entire hippocampus was applied to both labelmaps. This mask was defined as the voxels for which both J.D. and O.K. had labeled any subfield. Dice overlap was calculated across all labeled coronal slices as well as in unfolded space (Appendix 2), and for a full visualization of both raters' labels on all slices after matching gray matter bounds, see Appendix 2, *Supplementary file 1*.

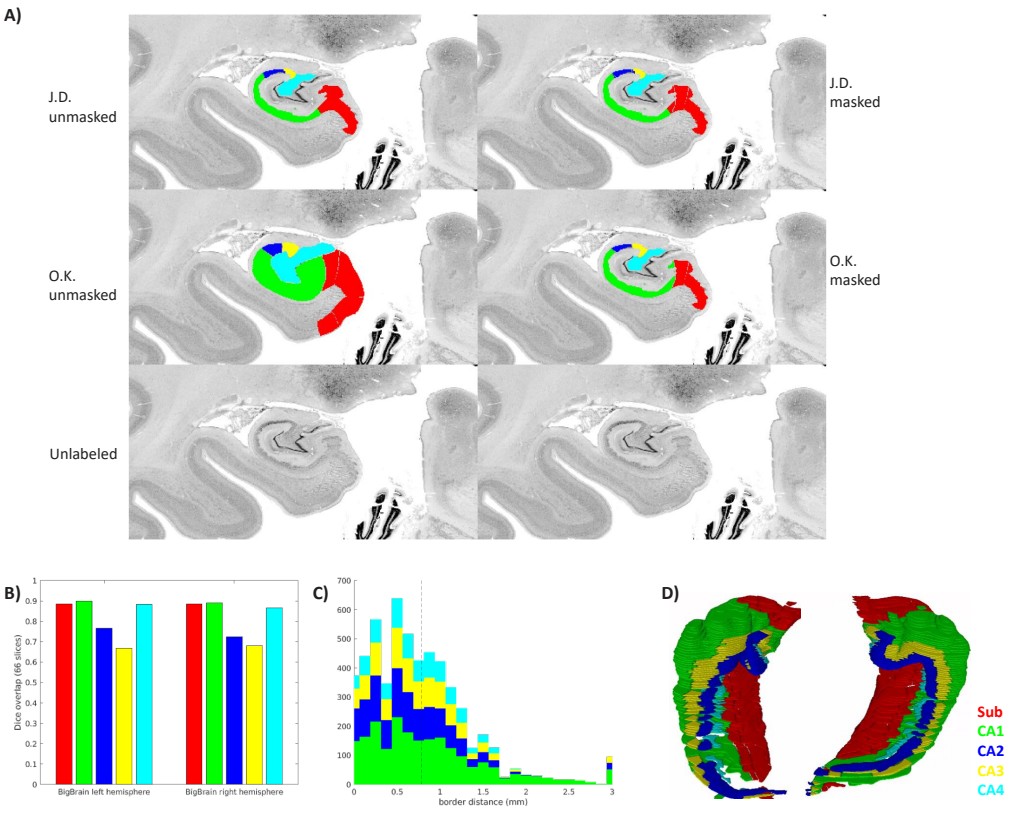

**Appendix 2—figure 1.** Inter-rater ground-truth subfield segmentation in the left and right BigBrain hippocampus (sample1 and sample2). (**A**) Gray matter mask (defined as any unlabeled voxel from either rater) applied to one slice. (**B. C**) show inter-rater Dice and border distances, respectively, as in *Figure 3*. (**D**) Stacked coronal slices from rater O.K.

The median distance between ground-truth raters was 0.781 µm, and Dice scores in coronal slices were in the range considered very good (0.8–0.9) for larger subfields Sub, CA1, and CA4, and moderate (0.6–0.8) for smaller subfields CA2 and CA3. It is interesting to note that in CA3 and CA4, the proposed unfolded registration method actually outperformed ground-truth label reliability. Ground-truth reliability should represent the ceiling for automated registration performance. Two factors, thus, likely led to systematically higher Dice scores and lower border distances in the unfolded registration evaluation: (1) all labeling was performed by rater J.D. who likely had different but consistent criteria for identifying borders, and (2) unfolded registrations are bound by the same distal 'endpoint.' This is defined within HippUnfold as being the DG or the true topological 'terminus' of the archicortex (making up also a portion of the true terminus of the cortex altogether). This terminus point can, thus, act as an anchor for homologous points between samples, which is reflected most strongly in the evaluation in overlap of the subfields closest to that point: CA4 and CA3. HippUnfold leverages this topological endpoint in determining homology between samples, but it is important to note that this method relies on consistent identification of that endpoint (which is relatively clear in histology but can be challenging in MRI, see *DeKraker et al., 2022* for discussion).

