## [Editor Report · eLife assessment]

This paper presents an **important** contribution to the field of hippocampal registration by introducing a novel surface-based approach that utilizes the topological and morphological features of the hippocampus for anatomical registration across individuals, rather than volumetric-based methods commonly used in the literature. The study provides **compelling** evidence for the efficacy of this approach using histological samples from three different datasets and offers validation of the method through comparison with traditional volumetric registration. This is **important** work given the large number of studies that examine hippocampal shape, thickness, and function in large cohorts, providing strong support for the use of hippocampal unfolding methods in future studies.

---

## [Referee Report · Reviewer #1 (Public Review)]

DeKraker et al. propose a new method for hippocampal registration using a novel surface-based approach that preserves the topology of the curvature of the hippocampus and boundaries of hippocampal subfields. The surface-based registration method proved to be more precise and resulted in better alignment compared to traditional volumetric-based registration. Moreover, the authors demonstrated that this method can be performed across image modalities by testing the method with seven different histological samples. This work has the potential to be a powerful new registration technique that can enable precise hippocampal registration and alignment across subjects, datasets, and image modalities.

---

## [Referee Report · Reviewer #2 (Public Review)]

Summary:

In the current manuscript, Dekraker and colleagues have demonstrated the ability to align hippocampal subfield parcellations across disparate 3D histology samples that differ in contrast, resolution, and processing/staining methods. In doing so, they validated the previously generated Big-Brain atlas by comparing across seven different ground-truth subfield definitions. This is an impressive effort that provides important groundwork for future in vivo multi-atlas methods.

Strengths:

DeKraker and colleagues have provided novel evidence for the tremendously complicated curvature/gyrification of the hippocampus. This work underscores the challenge that this complicated anatomy presents in our ability to co-register other types of hippocampal data (e.g. MRI data) to appropriately align and study a structure in which the curvature varies considerably across individuals.

This paper is also important in that it highlights the utility of using post-mortem histological datasets, where ground truth histology is available, to inform our rigorous study of the in vivo brain.

This work may encourage readers to consider the limitations of the current methods that they currently use to co-register and normalize their MRI data and to question whether these methods are adequate for the examination of subfield activity, microstructure, or perfusion in the hippocampal head, for example. Thus the implications of this work could have a broad impact on the study of hippocampal subfield function in humans.

Weaknesses:

As the authors are well aware, hippocampal subfield definitions vary considerably across laboratories. For example, some neuroanatomists (Ding, Palomero-Gallagher, Augustinack) recognize that the prosubiculum is a distinct region from subiculum and CA1 but others (e.g. Insausti, Duvernoy) do not include this as a distinct subregion. Readers should be aware that there is no universal consensus about the definition of certain subfields and that there is still disagreement about some of the boundaries even among the agreed upon regions.

---

## [Author Response]

The following is the authors’ response to the previous reviews

**Reviewer #2 (Public Review):**
DeKraker et al. propose a new method for hippocampal registration using a novel surface-based approach that preserves the topology of the curvature of the hippocampus and boundaries of hippocampal subfields. The surface-based registration method proved to be more precise and resulted in better alignment compared to traditional volumetric-based registration. Moreover, the authors demonstrated that this method can be performed across image modalities by testing the method with seven different histological samples. This work has the potential to be a powerful new registration technique that can enable precise hippocampal registration and alignment across subjects, datasets, and image modalities.

We thank the Reviewer, and feel this is an accurate summary of our work.

**Reviewer #3 (Public Review):**
Summary:In the current manuscript, Dekraker and colleagues have demonstrated the ability to align hippocampal subfield parcellations across disparate 3D histology samples that differ in contrast, resolution, and processing/staining methods. In doing so, they validated the previously generated Big-Brain atlas by comparing across seven different ground-truth subfield definitions. This is an impressive effort that provides important groundwork for future in vivo multi-atlas methods.Strengths:DeKraker and colleagues have provided novel evidence for the tremendously complicated curvature/gyrification of the hippocampus. This work underscores the challenge that this complicated anatomy presents in our ability to co-register other types of hippocampal data (e.g. MRI data) to appropriately align and study a structure in which the curvature varies considerably across individuals.This paper is also important in that it highlights the utility of using post-mortem histological datasets, where ground truth histology is available, to inform our rigorous study of the in vivo brain.This work may encourage readers to consider the limitations of the current methods that they currently use to co-register and normalize their MRI data and to question whether these methods are adequate for the examination of subfield activity, microstructure, or perfusion in the hippocampal head, for example. Thus the implications of this work could have a broad impact on the study of hippocampal subfield function in humans.Weaknesses:As the authors are well aware, hippocampal subfield definitions vary considerably across laboratories. For example, some neuroanatomists (Ding, Palomero-Gallagher, Augustinack) recognize that the prosubiculum is a distinct region from subiculum and CA1 but others (e.g. Insausti, Duvernoy) do not include this as a distinct subregion. Readers should be aware that there is no universal consensus about the definition of certain subfields and that there is still disagreement about some of the boundaries even among the agreed upon regions.

We thank the Reviewer, and feel this is an accurate summary of our work that also provides useful scientific context.

**Reviewer #2 (Recommendations For The Authors):**
The authors have done a great job with the revisions and have addressed all my concerns. They have clarified aspects of the method and procedure and have included a helpful walk-through explanation of an example subject. The authors have also expanded the discussion and addressed the motivation and justification for certain steps of the procedure.

We thank the Reviewer.

**Reviewer #3 (Recommendations For The Authors):**
The authors have addressed my previous comments and I believe the impact and take home message of the paper is more clear.

We thank the Reviewer.

In Figure 1, is the proximal-distal label reversed for panel B? I think P (proximal) should be closer to CA4/DG and D (distal) should be closer to subiculum. Am I misreading the graph?

We thank the Reviewer for this consideration, but the label is as intended. The terms proximal/distal in the hippocampal literature are sometimes relative to the dentate gyrus and sometimes relative to the rest of the cortex. In our case, we use the terms relative to the neocortex, following Ding and Van Hoesen (2015). We have now added the following to clarify this point at the first use of these terms (p.5):

“The current work, however, defined this tessellation as a regular mesh grid in unfolded space consisting of 256×128 points across the anterior-posterior (A-P) and proximal-distal (P-D) (relative to the neocortex) axes of the unfolded hippocampus, respectively.”